# Association between Sedentary Behavior and Cognitive Performance in Middle-Aged and Elderly Adults: Cross-Sectional Results from ELSA-Brasil

**DOI:** 10.3390/ijerph192114234

**Published:** 2022-10-31

**Authors:** Pricilla de Almeida Moreira, Sheila Maria Alvim de Matos, Francisco José Gondim Pitanga, Luana Giatti, Sandhi Maria Barreto, Rosane Harter Griep, Maria da Conceição Chagas de Almeida, Carlos Antônio de Souza Teles Santos

**Affiliations:** 1Postgraduate Program in Collective Health, Instituto de Saúde Coletiva, Universidade Federal da Bahia, Salvador 40220-141, BA, Brazil; 2Department of Physical Education, School of Education, Universidade Federal da Bahia, Salvador 40220-141, BA, Brazil; 3Postgraduate Program in Public Health, School of Medicine & Clinical Hospital, Universidade Federal de Minas Gerais, Belo Horizonte 31270-901, MG, Brazil; 4Laboratory of Health and Environment Education, Oswaldo Cruz Institute, Fundação Oswaldo Cruz, Rio de Janeiro 21040-360, RJ, Brazil; 5Gonçalo Moniz Institute, Fundação Oswaldo Cruz, Salvador 21045-900, BA, Brazil

**Keywords:** sedentary behaviour, performance, cognition, aging, ELSA-Brasil

## Abstract

Background: This study aimed to verify the association between Sedentary Behavior (SB) and performance on cognitive function tests in middle-aged and elderly adults. Methods: This cross-sectional study included 6505 participants (55.2% women) of ELSA-Brasil, with a median age of 61 years. The different types of SB considered were sitting time and screen time. The scores obtained in the memory, language, and executive function tests were used to assess cognitive performance (CP). The association between SB and CP was assessed using linear regression. Results: For men, sitting time was associated with better performance in memory, language, and executive function tests. Screen time on the weekend, showed a favorable association with performance in the executive function test. Occupational screen time on weekdays was positively associated with language test performance. For women, sitting time and occupational screen time were positively associated with performance on memory tests. SB was favorably associated with performance in language tests and executive function tests. Conclusions: SB seems to favor CP in this population without evident dementia and with a high level of education. The type of SB (mentally active or passive) and the schooling seem to be of particular interest for cognitive performance.

## 1. Introduction

Neurological disorders are among the main causes of disability worldwide [1]. Dementia affects more than 50 million people, of which more than 10 million are in the Americas region [2]. In Latin America, dementia affects approximately 7.1% of the elderly, a rate similar to that of high-income countries but with a higher prevalence in younger age groups (65–69 years) [3]. Approximately 60% of individuals with dementia live in low- and middle-income countries, with projections increasing to 70% by 2050 [4]. The impact of dementia includes physical, psychological, social, and economic aspects, affecting society as a whole [2]. Given this scenario, it is important to identify modifiable risk factors, since there is no curative treatment for dementia. Thus, the onset or progression of dementia can be delayed, and its global burden can be reduced [5].

Previous studies on a Brazilian cohort of adult and elderly workers identified several factors associated with performance on cognitive tests in this population, which were used as a proxy for cognitive function. Early exposure to adverse social conditions [6], the health conditions, such as the presence of diabetes mellitus [7], hypertension and pre-hypertension [8], low vascular health score [9], migraine [10], and low muscle strength [11] are some of the factors that showed an unfavorable association with cognitive performance in tests assessing memory, language, and executive function.

Among the modifiable risk factors associated with dementia, the role of lifestyle is well recognized and is considered a relevant starting point for the prevention, delay, or reduction in cognitive decline [12]. Regular and prolonged physical activity is associated with a reduced risk of cognitive decline and dementia [13]. However, current literature emphasizes the relationship between sedentary behavior and cognitive performance [14]. Sedentary behavior (SB) is not the opposite of physical inactivity but is defined as any waking behavior characterized by an energy expenditure ≤1.5 metabolic equivalents, while in a sitting, reclining, or lying position [15]. Emerging literature on the relationship between SB and cognitive function has shown inconsistent results. Although a systematic review [14] suggested an unfavorable association, prospective studies of adults and older adults without dementia found no evidence of an association between SB and global cognitive decline over a mean follow-up interval of 2.0–8.1 years [16]. These controversies seem to be related to the profile of the sample studied and the type of sedentary behavior assessed (sitting time or screen time).

Therefore, further studies are needed, especially in low- and middle-income countries where evidence of this association has not been identified and where the prevalence of dementia and the rate of population aging are increasing rapidly [17]. In addition, low- and middle-income countries also have a distinct socio-demographic profile, such as lower levels of education, greater miscegenation, and a high prevalence of chronic non-communicable diseases (NCDs), which are known risk factors for dementia [18,19].

Thus, this study aimed to verify the association between different types of sedentary behavior and performance on cognitive function tests in middle-aged and elderly adults. The elucidation of this association could help develop strategies to reduce the prevalence of dementia.

## 2. Materials and Methods

### 2.1. Study Design

This is a cross-sectional study with data from the first follow-up (wave 2:2012–2014) of the ELSA-Brasil. The ELSA-Brasil was a multicenter cohort study launched in the year 2008 in six Brazilian capitals; the participants were 15,105 retired and active civil servants aged between 35 and 74 years. More details regarding the study design and cohort profile can be found in previous publications [20,21]. ELSA-Brasil was approved by the Research Ethics Committees of the participating institutions and by the National Research Ethics Committee (CONEP 976/2006). All the participants provided written informed consent.

### 2.2. Sample

Among the 14,014 participants in the first follow-up (wave 2:2012–2014) of the ELSA-Brasil, 7248 individuals aged >55 years were eligible for this analysis because they had performed cognitive function tests. Of these, we excluded: (1) those with incomplete information for outcomes (n = 533), (2) those who reported having had a stroke (n = 113), (3) and those with incomplete data on the questionnaire on sedentary behavior (n = 97). The stroke patients were excluded due to possible impact on performance in cognitive tests. The final sample consisted of 6505 participants (Figure 1). After exclusion, no participants were identified as using anticholinesterase drugs.

### 2.3. Response Variables

The response variables were total cognitive test scores that assessed memory, language, and executive function. The tests applied are part of the battery of neuropsychological tests Consortium to Establish a Registry for Alzheimer’s Disease (CERAD) [22,23] and are considered a proxy for cognitive function.

Memory: In assessing memory, tests were used to assess immediate and delayed memory, in addition to the ability to remember and distinguish correct words from a set of distractors. To examine immediate memory, the word list memory test was used [22,23]. In this test, the subject is presented with a list of ten unrelated words, and immediately afterwards, he must remember as many words as he can. This procedure is performed thrice. In the delayed memory exam, the evocation test [22,23] was used, which measures the ability to retain previously said words after a time interval and other activities. Recognition of the words presented, when mixed with a set of distractors [22,23], is also part of the memory test. This test measures the ability to retrieve information through a list containing elements from the original list and a series of distractors. The memory test score represents the sum of the number of correct words in the three tests applied.

Language: Tests of semantic verbal fluency (flora category) and phonemic fluency (letter A) were used to assess declarative and semantic memory and language [22,23]. In the assessment of semantic fluency, the participants were asked to name as many words as possible in the flora category (vegetables, flowers, trees, fruits, etc.). In phonemic fluency, participants were asked to name words starting with the letter A. The final score corresponds to the number of correct words in 60 s in the two tests applied.

Executive function: The track B test assesses attention, concentration, psychomotor speed, and mental flexibility (executive function) [24]. In this test, the participant must draw a line connecting numbers and letters as quickly as possible without removing the pencil from the paper. The task execution time is then noted. Trail Making Test Version A was used for training version B (Trail B). The score is the time (in seconds) taken to complete the Trail B test.

The higher the scores on the memory and language tests, the better the cognitive performance in these domains. On the other hand, the longer the time taken to complete Track B, the worse the participant’s performance. All tests were adapted and validated for the Brazilian population [22]. Previous results from ELSA-Brasil showed that the memory tests had moderate reliability, the language tests had good reliability, and the Track B test had almost perfect reliability [25]. All tests were performed in a quiet room by trained interviewers using standardized protocols, and recorded and reviewed for quality control [26,27].

### 2.4. Exposure Variables

Sitting time: Participants were asked to report the average daily number of cumulative hours spent sitting on weekdays (Monday to Friday) and weekends (Saturday-Sunday) with the following questions: (a) How much time per day, on average, do you spend sitting during weekdays? (b) How much time per day, on average, do you spend sitting during the weekend?

Screen Time: Participants were asked to report the average daily number of accumulated hours spent sitting in front of a screen (TV, video games, computers, tablets, smartphones, etc.).

Thus, for the purposes of analysis, the following dimensions of sedentary behavior were considered: (1) time sitting during weekdays, (2) weekend sitting time, (3) leisure screen time during weekdays, (4) leisure screen time on the weekend, (5) weekday occupational screen time, and (6) weekend occupational screen time.

### 2.5. Co-Variables

Covariates in this study were self-reported using standardized questionnaires, or obtained through clinical procedures or standardized laboratory test measurements. Details of the procedures can be found in the literature [20,28].

Sociodemographic characteristics: Sex, race/skin color, marital status, and education were included in the study. Schooling was divided into four subgroups: <8 years (incomplete elementary school), 8 years (complete elementary school), 9–11 years (high school), and ≥12 years (higher education). Participants’ perception of the characteristics of the neighborhood (“environment conducive to physical activity” and “safety”) [29] was also evaluated. The higher the score, the better the assessment of the individual in relation to the characteristics of his neighborhood.

Socioeconomic position: It was evaluated according to the participant’s current socio-occupational class based on three aspects: (1) occupation, (2) expected income considering the educational level (average market value), and (3) reported income. The scores were summarized into three categories: high, medium, and low [30].

Occupation: Occupations were organized into three categories: academic, administrative, and operational activities, as proposed by Machado et al. [31]. Activity status (active or retired) was also assessed.

Health behaviors: Health behaviors included smoking (non-smoker, ex-smoker, and smoker), excessive alcohol consumption (≥210 g/week for men and ≥140 g/week for women), sleep (self-reported daily hours), and leisure-time physical activity. The practice of physical activity (PA) was measured using the International Physical Activity Questionnaire (IPAQ) [32] in the leisure PA domain. For analysis purposes, PA was categorized into: (1) active—individuals meeting any of the following criteria: (a) ≥150 min per week of moderate-intensity physical activity or walking and/or (b) ≥60 min per week of vigorous-intensity physical activity, or (c) ≥150 min per week of any combination of walking, moderate-intensity, or vigorous-intensity physical activity; (2) insufficiently active—individuals who do not meet the criteria to be classified as active or who do not report physical activity.

Health indicators: Anthropometric nutritional status was evaluated by body mass index (kg/m^2^), diagnosis of diabetes (self-reported or meeting any of the following criteria: use of anti-diabetic drugs, fasting glucose ≥126 mg/dL, glucose tolerance test ≥200 mg/dL, or glycated hemoglobin ≥6.5%), and hypertension (defined as systolic pressure >140 mmHg and/or diastolic pressure >90 mmHg and/or confirmed drug treatment for hypertension). Biochemical tests and weight check were performed under fasting conditions. The presence of depressive symptoms (at least two symptoms from Section G of the CIS-R) [33] and self-perception of health were also considered.

### 2.6. Statistical Analysis

Data were expressed as absolute and relative frequencies and measures of central tendency (mean or median) and dispersion (standard deviation or interquartile range). Data normality was tested using histograms and the Kolmogorov–Smirnov test. Categorical variables were compared using Pearson’s chi-square test. Mann–Whitney and Student’s *t*-tests were used to verify the difference between the numerical variables, according to their normality. For variables with observed and expected frequencies less than five and minimum expected frequencies greater than or equal to 1, the likelihood ratio was used. For the variable referring to executive function, logarithmic transformation was applied to base 10 to make it approximately normal. As SB and performance on cognitive tests differed significantly between men and women, the analyses were stratified by sex. Single and multiple linear regressions were used to obtain associations between sedentary behavior domains (sitting time on weekdays and weekends; leisure and occupational screen time, weekdays, and weekends) and test performance. Cognitive (memory, language, and executive function). Model 1 had simple associations between exposures and outcomes of interest; Model 2 had age-adjusted associations; and Model 3 had associations adjusted for variables associated with some of the outcomes (*p* < 0.05), in addition to those considered a priori as potential confounding factors (age, education, and physical activity). Multicollinearity was verified in Model 3 by determining the variance inflation factors (VIF), considering values of VIF >10 as multicollinearity problems. It is noteworthy that the variables “occupation” and “socioeconomic position” presented multicollinearity with each other. The first was prioritized to the detriment of the second because the latter comprises aspects related to occupation and income, which could cause other subsequent multicollinearity problems. Tests to evaluate assumptions of normality, linearity, and homoscedasticity confirmed the adequacy of the final model (Model 3). Statistical analyses were performed using *Stata* software (Stata Corporation, College Station, TX, USA) version 14. Statistical significance was set at *p* < 0.05.

## 3. Results

### 3.1. Characteristics of the Sample

Table 1 shows the characteristics of the study sample according to sex. Among the 6505 selected, 55.2% were female. The median age was 61 years (IQR 58–66). Most participants had completed higher education (58.2%), were racially white (56.1%) (and 40% were black and mixed race), had high socioeconomic status (41%), were married or in stable union (60.5%), were in academic occupations (43.6%), and had active occupational status (77.7%). Regarding health status, there was a high percentage of hypertensive individuals (51.8%), followed by diabetics (15.2%), individuals having depressive symptoms (13.3%), and overweight individuals (median = 26.98 kg/m^2^; IIQ:24.33–30.21 kg/m^2^). Nonetheless, most participants had a good or very good self-perception of health (80%). Regarding health behaviors, most participants did not report excessive alcohol intake (91.6%), were non-smokers (79.4%), were physically active (50.2%), and had a median of 6 h of sleep per day. Regarding performance on cognitive tests, the median score on the memory tests was 38 (IIQ 33–42), with a mean score of 28.8 (SD = 8.28) on the language tests, and a median time of 109 s (IIQ 81–157) in performing trial test part B. In the various aspects of sedentary behavior evaluated, it was observed that most of the sedentary time occurred during weekend which included time spent sitting (median = 4 h; IIQ 2.5–6) and leisure screen time (mean = 5.54 h; SD = 3.29).

Significant differences were observed according to sex for almost all investigated variables. It is noteworthy that women had a higher level of education (55.7% with complete higher education), were more active (52.7%), performed better in cognitive tests, and spent less sitting and screen time than men (Table 1).

For both sexes, the higher the level of education and the higher the socioeconomic position, the higher the average score in the memory and language domains, and the lower the average time on the Part B trial. At all levels of education and socioeconomic position, women performed better on cognitive tests than men, except for the execution time of the trial test, where men of a higher socioeconomic position performed slightly better than women in the same position. It was also observed that active individuals performed better on the cognitive tests (Appendix A).

### 3.2. Associations between Sedentary Behaviour and Cognitive Performance

Table 2 shows the results of the crude and adjusted associations between sedentary behavior and performance on cognitive tests. There was a statistically significant positive association between most types of sedentary behavior and performance on the memory test in both sexes, both in the crude and age-adjusted analyses. However, after adjusting for confounding factors (Model 3), this association only remained significant for sitting time in the week (β = 0.12, 95%CI 0.05; 0.20, men; β = 0.12, 95%CI 0.06; 0.17, women) and weekend (β = 0.10, 95%CI 0.03; 0.17, *p* value 0.007 for men), and for occupational screen time during weekdays (β = 0.10, 95%CI 0.03; 0.17, men; β = 0.06, 95%CI 0.00; 0.12 for women). Thus, after adjusting for potentially confounding variables, for every one-hour increase in sitting time during weekdays, there was a 0.12 increase in memory test scores for both men and women.

Likewise, there was a statistically significant positive association between sedentary behavior and performance in language tests in both sexes. After adjusting for confounders (Model 3), in men, this association remained significant for weekdays (β = 0.22, 95%CI 0.13; 0.31) and weekends (β= 0.13, 95%CI 0.04; 0.23). In women, these associations remained significant for sitting time during weekdays (β = 0.23, 95%CI 0.15; 0.31), on the weekend (β = 0.21, 95%CI 0.11; 0.30), leisure screen time on the weekend (β = 0.13, 95%CI 0.01; 0.25), occupational screen time during weekdays (β = 0.15, 95%CI 0.06; 0.24) and on the weekend (β = 0.17, 95%CI 0.02; 0.33). Thus, for each one-hour increase in sitting time during weekdays, for example, there was an increase of 0.22 and 0.23 in the language test scores for men and women, respectively, after adjusting for factors of confusion.

Regarding executive function, there was a significant inverse association with most sedentary behaviors. Thus, sitting time and screen time were associated with a shorter time to perform the Part B trial test. In men, sitting time during weekdays (β = −0.22, 95%CI −0.03; −0.00), on the weekend (β = −0.01, 95%CI −0.02; −0.01), leisure screen time on the weekend (β = −0.01, 95%CI −0.02; −0.00), occupational screen time during weekdays (β = −0.22, 95%CI −0.02; −0.01), and on the weekend (β = −0.01, 95%CI −0.02; 0.00) were associated with better performance on executive function tests, even after adjusting for confounding factors (Model 3). In women, there was also a significant inverse association between executive function and all sedentary behaviors analyzed (Model 3): sitting time during weekdays (β = −0.02, 95%CI −0.03; −0.02) and weekends (β = −0.02, 95%CI −0.03; −0.01), leisure screen time during weekdays (β = −0.01, 95%CI −0.02; −0.00) and on weekends (β = −0.02, 95%CI −0.03; −0.01), and occupational screen time during weekdays (β = −0.02, 95%CI −0.02; −0.01) and on the weekend (β = −0.01, 95%CI −0.02; −0.00).

## 4. Discussion

The results of the present study, based on a sample composed of adults and the elderly, from a middle-income country, without diagnosed dementia or previous stroke history, and with a high level of education, show a positive association between sedentary behavior and performance in cognitive tests measuring memory, language, and executive function. This is the first study to investigate this relationship in a population of adults and older adults in a middle-income country in South America.

For men, sitting time (during weekdays and weekend) was associated with better performance on memory, language, and executive function tests. Weekday occupational screen time was favorably associated with performance on memory and executive function tests. Weekend screen time (occupational and leisure) was associated with better performance on the executive function test. For women, sitting time and occupational screen time, both during weekdays, were positively associated with performance in memory tests. In terms of performance on language tests, all types of sedentary behavior analyzed were favorably associated with it, except screen time during leisure time in the week. Likewise, a positive association was observed between executive function tests and all types of sedentary behavior.

The association between sedentary behavior and cognitive function remains inconclusive. Some studies have identified unfavorable associations [17], others favorable [34,35], and some have identified no association [36,37]. In two recent systematic reviews on sedentary behavior and cognitive performance in young and middle-aged adults in the workplace [38] and community-dwelling elderly people without dementia [39], evidence suggests inconsistent associations between varying sedentary behaviors (sitting time, TV, and others, measured objectively and subjectively) and different domains of cognitive function. A secondary analysis of five cohorts found no cross-sectional or longitudinal association between sedentary behavior (assessed subjectively and objectively) and global cognition in the elderly [19].

Our hypothesis is that the divergences identified in the literature can be explained by the different types of sedentary behavior and cognitive performance evaluated and by the characteristics of study participants.

Different types of sedentary behavior seem to have different impacts on each cognitive skill assessed [17,40]. A large prospective cohort with data from the UK Biobank involving half a million adults showed that while watching television and driving was unfavorably associated with cognitive decline in all domains assessed (prospective memory, visual-spatial memory, numerical short-term memory, and fluid intelligence test), leisure computer use was associated with better performance in all of them [40]. There is also evidence that higher amounts of time spent using computers and lower amounts of time watching television may be related to better cognitive performance (verbal memory and executive function) in healthy older adults [34]. In a systematic review, Falck et al. [17] concluded that sedentary behavior is associated with worse cognitive performance based on the evaluation of eight studies, but only three of them considered computer use; the others, for the most part, evaluated based on time watching television. Thus, the findings should not be generalized but should be directed to each type of sedentary behavior and the respective cognitive abilities assessed.

Additionally, a recent study has suggested a classification of sedentary behavior [41]. According to the authors, sedentary behaviors can be divided into two categories: (1) mentally active, those that involve a greater amount of mental activity, and (2) mentally passive, those that are mentally passive in nature. Thus, although seated, individuals may be performing mentally active activities, such as participating in work-related meetings, performing tasks, and solving problems that require a certain mental “effort”, which can stimulate cognition [40,41]. Thus, categorization of sedentary behavior seems important as each category can influence health outcomes differently [41]. Our study population was mostly composed of skilled and active workers, predominantly involved in academic and administrative activities, which can be considered cognitively stimulating, contributing, in part, to the explanation of the favorable association between sedentary behavior and cognitive performance in this sample.

A cross-sectional study examined the association between sitting cognitive activity (CAS) and cognitive impairment in 5300 community-dwelling Japanese elderly individuals (mean age 75 years, SD 5.1) without dementia, disability, and severe illness [42]. The authors found that involvement in CAS (such as reading, writing, solving crossword puzzles, playing board games, and using a computer) was associated with a lower odds ratio for cognitive impairment (memory, attention, executive function, and processing speed), even after adjusting for confounders such as physical activity. These findings suggest that cognitive activities, even those performed in the sitting position, have a positive impact on cognition. Thus, while the duration of time spent in sedentary behavior is relevant for chronic diseases, the type of sedentary behavior (mentally active or passive) seems to be of particular interest for cognitive outcomes [41,43].

Another reason that may explain the conflicting results in the literature is the characteristics of the study samples. Cognitive performance can be strongly influenced by socio-demographic characteristics such as age and education [44,45]. The relatively low mean age and high level of education of the participants in the present study, through mechanisms involving greater cognitive reserve [46], seem to justify their relatively good cognitive performance. Additionally, studies have shown a positive association between sitting time, education, and social position. These results have been observed in German adults [47], middle-aged Australian adults [48], and young Brazilian adults [49]. Thus, it is possible that the relationship observed between types of sedentary behavior and cognitive performance reflects the characteristics of the study sample.

It is noteworthy that performance on all cognitive tests was worse among men than among women. These sex differences are not always identified and seem to depend on the assessed cognitive domain [50]. In addition, sex hormones appear to influence cognitive performance (e.g., executive function and memory) [50]. However, in a cross-sectional study, there is no way to conclude this relationship. Considering that schooling was higher among women than men at all corresponding educational levels, this may be one of the explanations for the sex differences in performance on cognitive tests.

### Strengths and Limitations

Some limitations of the present study must be considered. First, due to its cross-sectional nature, inferences about causality are not possible. In addition, the association between SB and cognitive decline could not be investigated since the ELSA-Brasil baseline does not include data on sedentary behavior, which makes it impossible to perform a longitudinal analysis of this association and, consequently, to assess the chronic effects of SB on cognitive function. We cannot rule out errors and bias in self-reported information on sitting time and screen time, which may have been underestimated. However, although objective data on SB (e.g., accelerometers) are more robust, they do not provide information about the context of sedentary behavior performed [40]. It is also noteworthy that the ELSA-Brasil cohort is not representative of the Brazilian population, since it includes active and skilled workers, most of whom have high schooling and higher socioeconomic status compared to the rest of the Brazilian population; therefore, this study’s results should be extrapolated with caution. In addition, we did not specify sedentary behavior according to occupational status (i.e., retired or active). Although in this study 77.7% of the sample is active we consider that the SB of a person who works and a person who no longer works may differentially influence cognition, and it should be investigated further with a more representative sample of retirees. It is also important to highlight that we assess the total sitting time, without distinguishing activities performed while in the sitting position (e.g., reading, writing, driving, etc.). Although previous studies also evaluated the total sitting time instead of specific activities [19], we encourage future studies to analyze specific types of sedentary behavior, which seems to play a central role in cognitive performance assessment. Finally, it is likely that the high volume of sitting and screen time, which was favorably associated with cognition in this study, is associated with other adverse health consequences, such as cardiometabolic markers (obesity, low HDL cholesterol levels, hypertriglyceridemia, and hypertension), as previously verified for this population [51].

Despite these limitations, this study has several strengths. The ELSA is the largest epidemiological study in South America [52], with social and regional diversity, with a high percentage of black and brown participants (around 40%); in most studies, the samples are predominantly composed of individuals of white skin [40,47,48]. In addition, we used validated and standardized tests applied by trained interviewers and reviewed them by a quality control team. It is also noteworthy that SB was evaluated in different types (sitting time and screen time) and contexts (leisure and occupational), as well as different cognitive skills (memory, language, and executive function). Data analysis was performed according to sex and used statistical models adjusted for a comprehensive set of potential confounders.

Finally, there are some recommendations for future studies on this topic. To date, most studies have separately investigated the effect of each physical behavior (sleep, physical activity, and sedentary behavior) on health outcomes, disregarding the co-dependent nature of these data [53]. Compositional data analysis (CoDA) has emerged as a way to overcome the limitations of conventional statistical models, considering the contribution of each behavior together, as part of a whole, instead of focusing on an isolated exposure as classically these data are analyzed [53]. Thus, future studies using objective methods of assessment of sedentary behavior and physical activity and adopting this new analysis approach may contribute to clarifying the contribution of different physical behaviors to different cognitive abilities. It should also be considered that cognitive tests are applied as a proxy for the cognition of individuals; however, they may be insufficient to capture all the complexity of cognitive function, which points to the need to use different assessment strategies such as clinical-neurological assessment and imaging tests [27].

## 5. Conclusions

The current study adds new evidence to the emerging body of research on sedentary behavior, physical activity, and cognitive performance. We identified that different types of sedentary behavior showed a favorable relationship with performance on cognitive tests assessing memory, language, and executive function in a sample of adults and elderly people without evident dementia and with high schooling. Our findings underscore the need to examine, in addition to the amount of sitting and screen time, the type of sedentary behavior and the context in which it is performed.

It is plausible that mentally active sedentary behaviors (e.g., reading and working while sitting) may stimulate and favor cognitive performance. However, before promoting the benefits of sedentary behavior on cognitive performance, more studies are needed, especially prospective studies, considering more aspects of cognition and sedentary behavior, employing objective methods such as accelerometers and brain imaging, and more data analysis approaches. In this way, one can gain a better understanding of the nature and magnitude of these associations. Given that this area of research is still under development and that results in middle-income countries are scarce, our study provides information that may contribute to a better understanding of the association between sedentary behavior and performance on cognitive tests.

We reiterate the importance of practicing physical activity and reducing the amount of time spent in sedentary behavior in accordance with current public health recommendations to promote the general health and well-being of the population.

## Figures and Tables

**Figure 1 ijerph-19-14234-f001:**
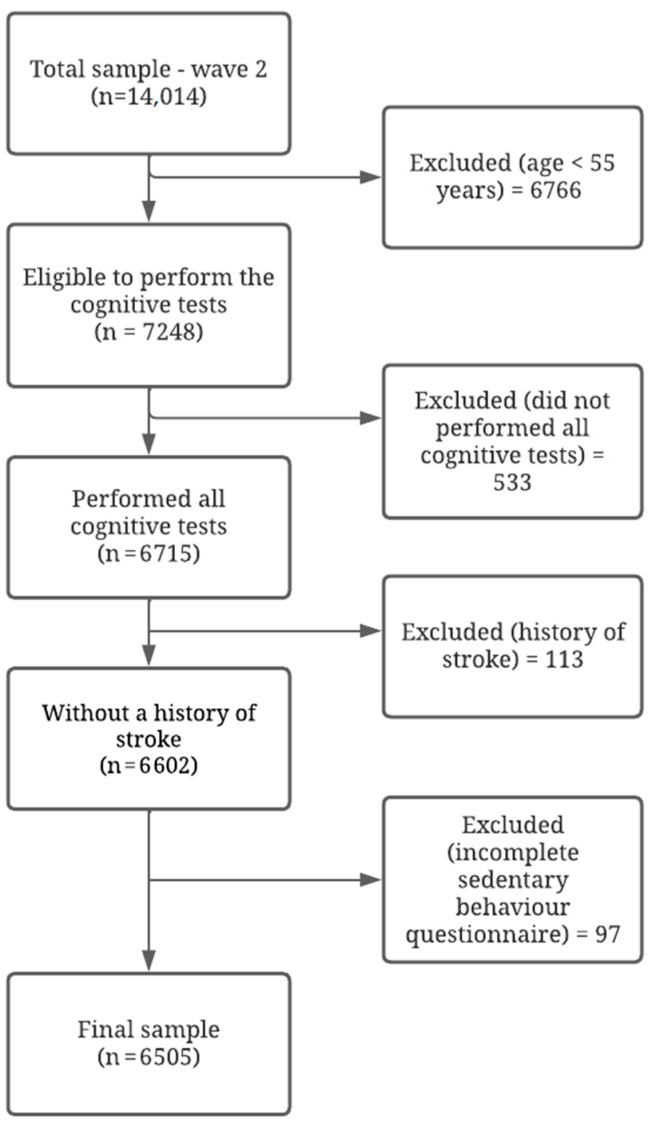
Flowchart of the study sample selection process.

**Table 1 ijerph-19-14234-t001:** Characterization of the sample stratified by sex. ELSA-Brasil (2012–2014).

	Total	Men	Women	*p* Value
Age (years)	n = 6505	n = 2915	n = 3590	0.001 ^‡^
	61 (58–66) ^¥^	62 (58–67) ^¥^	61 (58–66) ^¥^	
Schooling, n (%)	n = 6505	n = 2915	n = 3590	
Incomplete elementary school	355 (5.5%)	215 (60.6%)	140 (39.4%)	<0.001 ^£^
Complete elementary school	517 (7.9%)	263 (50.9%)	254 (49.1%)	
High school	1846 (28.4%)	759 (41.1%)	1087 (58.9%)	
Higher education	3787 (58.2%)	1678 (44.3%)	2109 (55.7%)	
Race/skin colour, n (%)	n = 6415	n = 2866	n = 3549	
Black	923 (14.4%)	344 (37.3%)	579 (62.7%)	<0.001 ^£^
Brown	1633 (25.5%)	737 (45.1%)	896 (54.9%)	
White	3597 (56.1%)	1685 (46.8%)	1912 (53.2%)	
Yellow	192 (3.0%)	69 (35.9%)	123 (64.1%)	
Indigenous	70 (1.1%)	31 (44.3%)	39 (55.7%)	
Socioeconomic position, n (%)	n = 6381	n = 2868	n = 3513	
High	2615 (41.0%)	1335 (51.1%)	1280 (48.9%)	<0.001 ^£^
Medium	2256 (35.4%)	792 (35.1%)	1464 (64.9%)	
Low	1510 (23.7%)	741 (49.1%)	769 (50.9%)	
Marital status, n (%)	n = 6505	n = 2915	n = 3590	
Married/stable union	3935 (60.5%)	2371 (60.3%)	1564 (39.7%)	<0.001 ^£^
Divorced	1157 (17.8%)	320 (27.7%)	837 (72.3%)	
Widower	515 (7.9%)	66 (12.8%)	449 (87.2%)	
Single	857 (13.2%)	142 (16.6%)	715 (83.4%)	
Other	41 (0.6%)	16 (39.0%)	25 (61.0%)	
Occupation, n (%)	n = 6381	n = 2868	n = 3513	
Academic	2783 (43.6%)	1357 (48.8%)	1426 (51.2%)	<0.001 ^£^
Administrative	2667 (41.8%)	894 (33.5%)	1773 (66.5%)	
Operational	931 (14.6%)	617 (66.3%)	314 (33.7%)	
Activity status, n (%)	n = 4218	n = 2085	n = 2133	
Active	3279 (77.7%)	1667 (50.8%)	1612 (49.2%)	<0.001 ^£^
Retired	939 (22.3%)	418 (44.5%)	521 (55.6%)	
Hypertension, n (%)	n = 6498	n = 2910	n = 3588	
No	3158 (48.6%)	1254 (39.7%)	1904 (60.3%)	<0.001 ^£^
Yes	3340 (51.4%)	1656 (49.6%)	1684 (50.5%)	
Diabetes type 2, n (%)	n = 6480	n = 2898	n = 3582	
No	5498 (84.8%)	2392 (43.5%)	3106 (56.5%)	<0.001 ^£^
Yes	982 (15.2%)	506 (51.5%)	476 (48.5%)	
Depressive symptoms, n (%)	n = 6504	n = 2915	n = 3589	
No	5638 (86.7%)	2655 (47.1%)	2983 (52.9%)	<0.001 ^£^
Yes	866 (13.3%)	260 (30.0%)	606 (70.0%)	
BMI (kg/m^2^)	n = 6485	n = 2908	n = 3577	0.002 ^‡^
	26.98 (24.33–30.21) ^¥^	26.81 (24.48–29.61) ^¥^	27.13 (24.19–30.76) ^¥^	
Self-perception of health, n (%)	n = 6501	n = 2912	n = 3589	
Very good	1858 (28.6%)	751 (40.4%)	1107 (59.6%)	<0.001 ^£^
Good	3343 (51.4%)	1581 (47.3%)	1762 (52.7%)	
Regular	1190 (18.3%)	542 (45.5%)	648 (54.5%)	
Bad	88 (1.4%)	34 (38.6%)	54 (61.4%)	
Very bad	22 (0.3%)	4 (18.2%)	18 (81.8%)	
Excessive alcohol consumption, n (%)	n = 6502	n = 2914	n = 3588	
No	5957 (91.6%)	2522 (42.3%)	3435 (57.7%)	<0.001 ^£^
Yes	545 (8.4%)	392 (71.9%)	153 (28.1%)	
Smoking, n (%)	n = 6504	n = 2914	n = 3590	
Never	3366 (51.8%)	1243 (36.9%)	2123 (63.1%)	<0.001 ^£^
Former	2491 (38.3%)	1367 (54.9%)	1124 (45.1%)	
Current	647 (9.9%)	304 (47.0%)	343 (53.0%)	
Physical Activity, n (%)	n = 6505	n = 2915	n = 3590	
Insufficiently active	3239 (49.8%)	1371 (42.3%)	1868 (57.7%)	<0001
Active	3266 (50.2%)	1544 (47.3%)	1722 (52.7%)	
Sleep (hours/day)	n = 6499	n = 2914	n = 3590	0.617 ^‡^
	6 (6–7) ^¥^	6 (6–7) ^¥^	6 (6–8) ^¥^	
	8 (7–10) ^¥^	8 (7–9) ^¥^	8 (7–10) ^¥^	
Cognitive performance—memory *	n = 6505	n = 2915	n = 3590	
	38 (33–42) ^¥^	36 (32–40) ^¥^	39 (35–43) ^¥^	<0.001 ^‡^
Cognitive performance—language **	n = 6505	n = 2915	n = 3590	
	28.28 (8.28) ^§^	26.98 (8.25) ^§^	29.33 (8.15) ^§^	<0.001 ^#^
Cognitive performance—executive function ***	n = 6505	n = 2915	n = 3590	
	109 (81–157) ^¥^	110 (81–164) ^¥^	108 (80–152) ^¥^	0.029 ^‡^
Sitting time—weekdays (hours/day)	n = 6505	n = 2915	n = 3590	
	2.44 (1.75) ^§^	2.47 (1.75) ^§^	2.41 (1.75) ^§^	<0.001 ^#^
Sitting time—weekends (hours/day)	n = 6505	n = 2915	n = 3590	
	4 (2.5–6) ^¥^	4 (3–6) ^¥^	4 (2–6) ^¥^	<0.001 ^‡^
Leisure screen time—weekdays (hours/day)	n = 6505	n = 2915	n = 3590	
	2.98 (2.08) ^§^	5.84 (3.30) ^§^	5.30 (3.27) ^§^	0.169 ^#^
Leisure screen time—weekends (hours/day)	n = 6505	n = 2915	n = 3590	
	5.54 (3.29) ^§^	3.11 (2.08) ^§^	2.87 (8.07) ^§^	<0.001 ^#^
Occupational screen time—weekdays (hours/day)	n = 6505	n = 2915	n = 3590	
	2 (0–6) ^¥^	3 (0.5–6) ^¥^	2 (0–6) ^¥^	<0.001 ^‡^
Occupational screen time—weekends (hours/day)	n = 6505	n = 2915	n = 3590	
	0 (0–2) ^¥^	1 (0–2) ^¥^	0 (0–2) ^¥^	<0.001 ^‡^

n: frequency; %: percentage; BMI: body mass index; DM: Diabetes Mellitus; BMI: Body Mass Index. ^§^ Mean (standard deviation); ^¥^ Median (IQR: interquartile range); ^£^ Pearson’s chi-square test; ^‡^ Mann–Whitney test; ^#^ Student’s *t* test for independent samples; statistical significance *p* < 0.05. * Learning, recall and word recognition: number of correct words: score ranges from 0 to 50. ** Semantic and phonemic verbal fluency: number of words remembered in 1 min. *** Trail test (part B): time (in seconds) to perform the test.

**Table 2 ijerph-19-14234-t002:** Cross-sectional associations between types of sedentary behavior and performance on cognitive tests according to sex. ELSA-Brasil (2012–2014) (n = 6505).

	Model 1		Model 2		Model 3	
	β (IC_95%_) ^a^	*p* Value	β (IC_95%_) ^a^	*p* Value	β (IC_95%_) ^a^	*p* Value
Men						
Memory *						
Sitting time (weekdays)	0.40 (0.33; 0.47)	<0.001	0.38 (0.31; 0.45)	<0.001	0.12 (0.05; 0.20)	0.001
Sitting time (weekends)	0.28 (0.21; 0.36)	<0.001	0.29 (0.21; 0.36)	<0.001	0.10 (0.03; 0.17)	0.007
Leisure screen time (weekdays)	−0.18 (−0.32; −0.05)	0.008	−0.15 (−0.29; −0.02)	0.022	−0.02 (−0.14; 0.11)	0.817
Leisure screen time (weekends)	0.05 (−0.06; 0.17)	0.357	0.02 (−0.09; 0.14)	0.663	0.04 (−0.07; 0.14)	0.498
Occupational screen time (weekdays)	0.50 (0.43; 0.58)	<0.001	0.45 (0.38; 0.52)	<0.001	0.10 (0.02; 0.18)	0.014
Occupational screen time (weekends)	0.35 (0.22; 0.47)	<0.001	0.31 (0.19; 0.43)	<0.001	−0.04 (−0.15; 0.07)	0.498
Language **						
Sitting time (weekdays)	0.71 (0.62; 0.80)	<0.001	0.70 (0.61; 0.79)	<0.001	0.22 (0.13; 0.31)	<0.001
Sitting time (weekends)	0.51 (0.41; 0.62)	<0.001	0.51 (0.41; 0.62)	<0.001	0.13 (0.04; 0.23)	0.006
Leisure screen time (weekdays)	−0.18 (−0.35; −0.01)	0.044	−0.16 (−0.33; 0.01)	0.060	0.06 (−0.09; 0.21)	0.422
Leisure screen time (weekends)	0.10 (−0.04; 0.24)	0.155	0.09 (−0.05; 0.23)	<0.001	0.09 (−0.03; 0.22)	0.152
Occupational screen time (weekdays)	0.72 (0.62; 0.82)	<0.001	0.71 (0.61; 0.82)	<0.001	0.01 (−0.10; 0.11)	0.924
Occupational screen time (weekends)	0.65 (0.49; 0.82)	<0.001	0.64 (0.47; 0.81)	<0.001	0.01 (−0.13; 0.15)	0.851
Executive ***						
Sitting time (weekdays)	−0.06 (−0.06; −0.05)	<0.001	−0.06 (−0.06; −0.05)	<0.001	−0.02 (−0.03; −0.02)	<0.001
Sitting time (weekends)	−0.04 (−0.05; −0.04)	<0.001	−0.04 (−0.05; −0.04)	<0.001	−0.01 (−0.02; −0.01)	<0.001
Leisure screen time (weekdays)	0.02 (0.01; 0.03)	0.004	0.02 (0.00; 0.03)	0.007	−0.00 (−0.01; 0.01)	0.819
Leisure screen time (weekends)	−0.01 (−0.02; −0.00)	0.012	−0.01 (−0.02; −0.00)	0.022	−0.01 (−0.02; −0.00)	0.004
Occupational screen time (weekdays)	−0.07 (−0.07; −0.06)	<0.001	−0.07 (−0.07; −0.06)	<0.001	−0.02 (−0.02; −0.01)	<0.001
Occupational screen time (weekends)	−0.06 (−0.07; −0.05)	<0.001	−0.06 (−0.07; −0.04)	<0.001	−0.01 (−0.02; 0.00)	0.076
Women						
Memory *						
Sitting time (weekdays)	0.37 (0.31; 0.43)	<0.001	0.32 (0.26; 0.38)	<0.001	0.12 (0.06; 0.17)	<0.001
Sitting time (weekends))	0.25 (0.19; 0.32)	<0.001	0.24 (0.17; 0.31)	<0.001	0.07 (−0.00; 0.13)	0.053
Leisure screen time (weekdays)	−0.03 (−0.13; 0.08)	0.645	0.04 (−0.07; 0.14)	0.497	0.08 (−0.02; 0.19)	0.110
Leisure screen time (weekends)	0.04 (−0.06; 0.13)	0.437	0.02 (−0.07; 0.11)	0.649	0.05 (−0.04; 0.14)	0.325
Occupational screen time (weekdays)	0.39 (0.33; 0.45)	<0.001	0.31 (0.25; 0.37)	<0.001	0.06 (0.00; 0.12)	0.048
Occupational screen time (weekends)	0.49 (0.39; 0.59)	<0.001	0.42 (0.32; 0.52)	<0.001	0.08 (−0.01; 0.18)	0.081
Language **						
Sitting time (weekdays)	0.65 (0.57; 0.73)	<0.001	0.63 (0.54; 0.71)	<0.001	0.23 (0.15; 0.31)	<0.001
Sitting time (weekends)	0.56 (0.46; 0.67)	<0.001	0.55 (0.45; 0.66)	<0.001	0.21 (0.11; 0.30)	<0.001
Leisure screen time (weekdays)	−0.05 (−0.21; 0.11)	0.553	−0.00 (−0.16; 0.15)	0.972	0.01 (−0.13; 0.16)	0.852
Leisure screen time (weekends)	0.16 (0.03; 0.30)	0.018	0.15 (0.02; 0.29)	0.026	0.13 (0.01; 0.25)	0.035
Occupational screen time (weekdays)	0.63 (0.55; 0.72)	<0.001	0.61 (0.52; 0.70)	<0.001	0.15 (0.06; 0.24)	0.001
Occupational screen time (weekends)	0.84 (0.68; 1.01)	<0.001	0.80 (0.63; 0.97)	<0.001	0.17 (0.02; 0.33)	0.027
Executive ***						
Sitting time (weekdays)	−0.05 (−0.06; −0.05)	<0.001	−0.05 (−0.05; −0.04)	<0.001	−0.02 (−0.03; −0.02)	<0.001
Sitting time (weekends)	−0.05 (−0.05; −0.04)	<0.001	−0.04 (−0.05; −0.04)	<0.001	−0.02 (−0.03; −0.01)	<0.001
Leisure screen time (weekdays)	−0.00 (−0.01; 0.01)	0.403	−0.01 (−0.02; −0.00)	0.050	−0.01 (−0.02; −0.00)	0.022
Leisure screen time (weekends)	−0.02 (−0.03; −0.01)	<0.001	−0.02 (−0.03; −0.01)	0.001	−0.02 (−0.03; −0.01)	<0.001
Occupational screen time (weekdays)	−0.05 (−0.06; −0.05)	<0.001	−0.05 (−0.05; −0.04)	<0.001	−0.02 (−0.02; −0.01)	<0.001
Occupational screen time (weekends)	−0.06 (−0.07; −0.05)	<0.001	−0.06 (−0.07; −0.05)	<0.001	−0.01 (−0.02; −0.00)	0.008

^a^ Beta coefficient and 95% confidence interval. Statistical significance *p* < 0.05. * Learning, recall and word recognition: number of correct words: score ranges from 0 to 50. ** Semantic and phonemic verbal fluency: number of words remembered in 1 min. *** Trail test (part B): time (in seconds) to perform the test. Model 1: crude; Model 2: age-adjusted; Model 3: adjusted for age, schooling, neighborhood, physical activity, occupational nature, skin color, marital status, smoking, alcohol consumption, hours of sleep, self-perception of health, Body Mass Index, arterial hypertension and diabetes mellitus.

## Data Availability

The dataset used during the current study is available from the corresponding author upon reasonable request.

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
