# Peer review of "Association between Sedentary Behavior and Cognitive Performance in Middle-Aged and Elderly Adults: Cross-Sectional Results from ELSA-Brasil"

_ijerph, 2022, doi:10.3390/ijerph192114234_

Round 1
Reviewer 1 Report
Dear authors
I hope everyone is doing well, and with good health.
The manuscript is interesting and an important paper. It was a pleasure to review this study.
The study has relevance in decision-making in public health and epidemiological surveys. In ethnate, we cannot rule out measurement errors and biases in self-reported information, especially regarding the variables of sitting time and screen time. I have just a few questions: - Is the study limited due to the time of analysis?, since only data were collected up to 2014, so the population profile may have changed over the years. - Was the power analyzed?, since the loss of sample limits the inferences. - With the adjusted analyses, was the statistical correction made for the inferences?. - Effect and power analysis can guide the interpretation of findings. - A principal component analysis would provide guidance for the main correlated variables.
Best wishes
Author Response
Dear reviewer,
We really appreciate your time to review the present article, and your relevant contributions in order to improve the paper.
Please, find our response to your comments below:
- Is the study limited due to the time of analysis?, since only data were collected up to 2014, so the population profile may have changed over the years.
I appreciate your observation. Indeed, the results referend to the period of the study. ELSA studies with data from wave 1 show that the cohort profile keeps similar over the years (please, check on doi: 10.1093/ije/dyu027).
- Was the power analyzed?, since the loss of sample limits the inferences.
We appreciate your observation. The power of the sample size was verified, for each stratum, male (n=2915; mean 26.98) and female (n=3590; mean 29.33), for the outcome Cognitive performance – language, obtaining the power > 99% for both strata (sampsi -- Sample size and power for means in STATA), at the hypothesized value Ho: m = 28.28, where m is the mean in the population with SD 8.28.
3. With the adjusted analyses, was the statistical correction made for the inferences?
We thank you for your observation. The beta measurements presented in model 2 and model 3, in relation to model 1, take into account the presence of the age covariate (model 1) and other covariates (model 3), i.e., model 2 and 3 was adjusted for confounding.- A principal component analysis would provide guidance for the main correlated variables.
We really appreciate your suggestion. Indeed, applying for a PCA could be interesting. However, these variables were used to adjust the main association and for this reason, because of the collinearity (VIF > 10), the prioritization of one to the detriment of the other did not interfere with the interpretability of the association under study.
Additionally, we opted to analyze how it is presented in this paper to facilitate comparison with other studies that had the same aim.
Many thanks for your time reviewing the response.
We are looking forward to hearing from you.
Reviewer 2 Report
This paper examined the relationship between sedentary behavior and cognitive performance in adult and elderly adults from Brasil. It found different activities when time spent sitting are associated with different aspects of cognitive performance; interestingly, men and women showed different associations.
I am concerned over the theoretical formation of this paper. Since the paper specify SB into leisure vs occupational activities, why not just name it specific activities rather than SB? Do people use computer, read, or write while being active? it seems people usually perform these activities while sitting, then why not make it more specific by calling out specific activity instead of giving it a term SB? This brings to my concern over the introduction. It doesn’t seem to lay a solid foundation for looking into the relationship between SB and cognitive performance. I don’t think the behavior “sitting” captures on the nature of these activities and the outcome of interest, which is on cognitive level. It’d make more sense to evaluate the cognitive nature of the activities that people engage, for example, video game may improve creativity, although people usually sit when they play video games, it’s not sitting makes them creative but playing the video game itself. The fact that SB does not capture the nature of the activities is a major concern over the theoretical formation of this paper.
Methods:
1. From Table 1, it seems most variables differ significantly between men and women. Given that the sample is big, is it possible to do propensity score matching to eliminate the difference between the groups? To increase the similarity between the groups.
2. Is there a reason to include model 1 and 2 since model 3 already took into account of cofounding variables?
Discussion: after reporting the main findings, it’d be important to discuss the difference in associations between men and women. Why would gender affect the association reported in the study? The paper went on to discuss the inconsistent findings in the literature and potential reasoning for the discrepancy, which would be better fit in the introduction than discussion.
Same goes for the categorization of SB. It’d be more relevant to introduce it upfront than in the discussion.
Minor:
1. the paper needs careful proofread. Several sentences are followed by #. E..g, line 380. “about the context of sedentary behaviour performed40”
2. when citing the literature, the authors can be more explicit thus readers don’t have to go back and look up the ref. for example, “sex hormones appear to influence cognitive performance” what are the cognitive performances?
Author Response
Dear reviewer,
We really appreciate your time to review the present article, and your relevant contributions in order to improve the paper.
Please, find our response to your comments below:
- I am concerned over the theoretical formation of this paper. Since the paper specify SB into leisure vs occupational activities, why not just name it specific activities rather than SB? Do people use computer, read, or write while being active? it seems people usually perform these activities while sitting, then why not make it more specific by calling out specific activity instead of giving it a term SB? This brings to my concern over the introduction. It doesn’t seem to lay a solid foundation for looking into the relationship between SB and cognitive performance. I don’t think the behavior “sitting” captures on the nature of these activities and the outcome of interest, which is on cognitive level. It’d make more sense to evaluate the cognitive nature of the activities that people engage, for example, video game may improve creativity, although people usually sit when they play video games, it’s not sitting makes them creative but playing the video game itself. The fact that SB does not capture the nature of the activities is a major concern over the theoretical formation of this paper.
We appreciate your observation. The field of SB is relatively new and the concepts are not yet consensually defined. But we highlight the theoretical background on which this study is based.
Firstly, when the ELSA wave 2 questionnaire was constructed (in 2011), we used the concept of Sedentary Behavior (SB) from Pate, O'Neill, Lobelo (2008): "Sedentary behavior refers to any activity characterized by low energy expenditure, not exceeding 1.5 metabolic equivalents" (please, check on DOI: 10.1097/JES.0b013e3181877d1a). This concept has been widely used since then, even though at that time, the literature on SB was emerging and mostly analyzed the time sitting.
Currently, most studies in the BS area adopt the theoretical definition proposed by the Sedentary Behavior Research Network (2017) (doi: https://doi.org/10.1186/s12966-017-0525-8): "SB is any activity in the sitting position, reclined with energy expenditure minus 1.5 METs". Therefore, the different activities with low energy expenditure that individuals perform in the sitting position (computer use, reading, writing, etc.) are called sedentary behaviors. Several international guidelines, including that of the WHO, use the term SB, referring to a wide range of activities with low energy expenditure while in a sitting position (Please, check on: WHO. Guidelines on physical activity and
sedentary behaviour. World Health Organization, 2020; Department of Health,
Australian Government. Physical activity and sedentary behaviour guidelines – adults (18 to 64 years) – fact sheet, 2014; Canadian 24-Hour Movement Guidelines for Adults aged 18-64 years: An Integration of Physical Activity, Sedentary Behaviour, and Sleep. 2020). Thus, the use of the term SB has a well-founded theoretical basis in the literature.
Second, the SB concept does not include the context in which it takes place (whether occupational or leisure) and specific activities performed. Some studies specifically ask about the activities performed (watching television, using the computer, driving, etc.), however, in our study the questions were related to sitting time and screen time (Line 132-144), not being analyzed which specific activities were performed while in the sitting position. Previous studies of this cohort have been published with these data (please check on: doi10.1177/2050312119827089 https://doi.org/10.12820/rbafs.23e0006) and other studies have also evaluated in this way (i.e., without capture the nature of the activities) (please check on: https://doi.org/10.1007/s40279-019-01186-7).
Finally, indeed, the analysis of the TIME sitting alone does not allow us to identify the nature of the activities carried out while sitting. On the other hand, we assessed SB within a context (occupational and leisure), so that we could potentially identify the nature of activities performed while seated. However, we consider it relevant that the nature of the activity is considered - this is what we highlighted in Lines 331-342. We are willing to highlight as a limitation of our study the fact that the nature of the activities performed while in the sitting position was not directly analyzed.
2. From Table 1, it seems most variables differ significantly between men and women. Given that the sample is big, is it possible to do propensity score matching to eliminate the difference between the groups? To increase the similarity between the groups.
We really appreciate your suggestion. Indeed, to apply a PSM would be quite interesting. However, we opted to analyze how it is presented in this paper to facilitate comparison with other studies that had the same aim, since this is how it is traditionally presented in this way in epidemiological studies. We consider that this econometric analysis is an alternative for another moment.
3. Is there a reason to include model 1 and 2 since model 3 already took into account of cofounding variables?
Thank you for your observation. Indeed, model 3 is sufficient to show our main results. The purpose of showing models 1 and 2 is to allow the reader to identify the changes that occur in the association between SB and CF when confounding variables are inserted (especially age and education). But all descriptions and interpretations of results were based on model 3. Furthermore, the form of data presentation follows the “model” of previous papers published in the Journal that present the results in this way (please, check on: https://doi.org/10.3390/ijerph191912614).
4. Discussion: after reporting the main findings, it’d be important to discuss the difference in associations between men and women. Why would gender affect the association reported in the study? The paper went on to discuss the inconsistent findings in the literature and potential reasoning for the discrepancy, which would be better fit in the introduction than discussion. Same goes for the categorization of SB. It’d be more relevant to introduce it upfront than in the discussion.
I really appreciate your consideration of the discussion. Discussing gender differences was challenging, as we identified few studies that analyzed this data by sex, to the best of our knowledge. Furthermore, the literature on cognitive performance in men and women is scarcely and the results are inconsistent. However, in lines 364-370 we discussed the difference observed.
Finally, we speak about the inconsistent findings in the literature in the introduction (Line 59-64), and how we categorized the SB in methods (Line 141-144).
5. the paper needs careful proofread. Several sentences are followed by #. E..g, line 380. “about the context of sedentary behaviour performed40”
Thank you for your observation. The manuscript was carefully reviewed.
6. when citing the literature, the authors can be more explicit thus readers don’t have to go back and look up the ref. for example, “sex hormones appear to influence cognitive performance” what are the cognitive performances?
We appreciate your suggestion. We provided some details in order to give more clarification.
---
Many thanks for your time reviewing the comments.
We are looking forward to hearing from you.
Best wishes
Reviewer 3 Report
Dear Authors,
The manuscript entitled "Association between sedentary behavior and cognitive performance in middle-aged and elderly adults: cross-sectional results from ELSA-Brasil” is an interesting and pertinent topic about aging, and the importance of knowing more about the aging evaluation to promote healthy aging.
However, the manuscript is confusing, and it is important to clarify some questions.
I suggest some changes to better understand the manuscript.
S1. Rephrase the keywords: screen time and sitting time
I would try to find some more appropriate words, for example: performance and aging or elderly.
Introduction
S2. Line 36-38
You are a bit confused… what do you understand about the elderly in Brazil?
65-69 are younger groups, not elderly. I suggest reviewing and clarifying this information.
S3. Line 38-40
This sentence has no continuity with what was said earlier. I would suggest making a paragraph and talking about the impact of dementia.
S4. Line 44
Brazilian or Brasilian?!
Brazil/Brasil?!
Standardise
S5. Line 46
I suggest clarifying this statement:
“Early exposure to adverse social conditions [6], the health conditions, such as the presence of diabetes mellitus [7], hypertension and pre-hypertension [8], low vascular health score [9], migraine [10], and low muscle strength [11] are some of the factors that showed an unfavourable association with cognitive performance in tests assessing memory, language, and executive function.”
Material and Methods
S6. It is necessary to explain better what is ELSA-Brasil.
Results
S7. It is important to clarify what sedentary behaviour is.
It is completely different from the sedentary behaviour of a person who works and a person who no longer works, we are comparing two different samples...
It is important to better characterize the sitting time, leisure screen time, and occupational screen time, with active and passive samples...
S8. Format table 2. Check
Discussion
S9.
It was important to better clarify the results for adults and the elderly... it becomes a bit confusing because these two samples with different particularities are fair.
It is important to separate the sample between adults and the elderly...
And to understand the sedentary lifestyle in each of these age groups, for a better understanding of the study objective
This question is not very clear and the crossing of the variables should be redone.
Then, in the discussion, there are comparisons with studies in which the age range is much higher, which does not justify your results.
S10. Also, the notion of sedentary behaviour should be presented in the introduction and not in the discussion.
Author Response
Dear reviewer,
We really appreciate your time to review the present article, and your relevant contributions in order to improve the paper.
Please, find our response to your comments below:
S1. Rephrase the keywords: screen time and sitting time. I would try to find some more appropriate words, for example: performance and aging or elderly.
Thank you for your suggestion. It was added to the manuscript.
S2. Line 36-38. You are a bit confused… what do you understand about the elderly in Brazil? 65-69 are younger groups, not elderly. I suggest reviewing and clarifying this information.
We appreciate your observation. We used the term “elderly” for this Brazilian sample, because under Brazilian law and the literature in aging field in Latino America, older adults are 60 years of age or older (please, check on DOI http://dx.doi.org/10.20947/S0102-3098a0129 https://doi.org/10.1371/journal.pone.0236280 ). In our study 63% of the participants are elderly (> 60y), so we considered it more appropriate to use the term “elderly”.
S3. Line 38-40. This sentence has no continuity with what was said earlier. I would suggest making a paragraph and talking about the impact of dementia.
Thank you for your suggestion. We provided a sentence talking about the impact of dementia (Line 40-41).
S4. Line 44
Brazilian or Brasilian?! Brazil/Brasil?!
We reviewed the article, and we notice that throughout the manuscript it is written "Brazil" and "Brazilian". However, the study name is written as ELSA-Brasil because it is standard for all publications from this cohort. That’s why when we mention the study’s name, we write it as Brasil and not as Brazil.
S5. Line 46
I suggest clarifying this statement: “Early exposure to adverse social conditions [6], the health conditions, such as the presence of diabetes mellitus [7], hypertension and pre-hypertension [8], low vascular health score [9], migraine [10], and low muscle strength [11] are some of the factors that showed an unfavourable association with cognitive performance in tests assessing memory, language, and executive function.”
We really appreciate your suggestion. It was added to the manuscript.
S6. It is necessary to explain better what is ELSA-Brasil.
Thank you for your suggestion. Lines 78-80 and 385-387 provide an explanation of ELSA-Brasil. Plus, 2 references about the study design and cohort profile were given.
S7. It is important to clarify what sedentary behaviour is.
It is completely different from the sedentary behaviour of a person who works and a person who no longer works, we are comparing two different samples... It is important to better characterize the sitting time, leisure screen time, and occupational screen time, with active and passive samples...
We thank you for your recommendation. We brought the SB definition in the introduction (LINE 56-58). However, to the best of our knowledge, there is no specific definition of SB according to activity status (i.e., retired or employed). In addition, SB was analyzed in leisure time (i.e., no occupational time), which includes the analysis of the SB of those who are not working. Finally, in this study 77.7% of the sample is Active. Considering these aspects, in this exploratory study, we did not aim to analyze these dates separately, even though we recognize it would be interesting for future studies with a more representative sample of retirees. However, we are willing to include it as a limitation of the present study if it is considered necessary.
S8. Format table 2. Check
Thank you for your suggestion. We recheck Table 2. Please, let us know if there is any specific suggestion.
S9. It was important to better clarify the results for adults and the elderly... it becomes a bit confusing because these two samples with different particularities are fair. It is important to separate the sample between adults and the elderly...
And to understand the sedentary lifestyle in each of these age groups, for a better understanding of the study objective. This question is not very clear and the crossing of the variables should be redone. Then, in the discussion, there are comparisons with studies in which the age range is much higher, which does not justify your results.
Thank you, it is a fair observation. We are aware of these particularities that’s why we performed these analyzes for another manuscript where we were able to focus on this specific aspect. The present study is the first study to investigate this relationship in a middle-income country in South America, so we focused to provide an exploratory analysis of the association between SB and cognitive performance by sex.
In regard to the comparisons with other studies, there is a limitation in the literature with studies assessing SB and cognitive performance in adults and the elderly. In addition, we had the care to present the mean age or age group of the sample of the studies we cited. Finally, the study we cited in Line 345 (mean age 75 years, SD 5.1) was used to support our suggestion that “cognitive activities, even those performed in the sitting position, have a positive impact on cognition”. Actually, it was not used to compare directly with the sample of the present manuscript.
S10. Also, the notion of sedentary behaviour should be presented in the introduction and not in the discussion.
Thank you for your observation. We present the notion of sedentary behavior in the introduction (please, check Line 56-58).
---
Many thanks for your time reviewing the comments. We appreciate your suggestions.
We are looking forward to hearing from you.
Best wishes
Round 2
Reviewer 2 Report
Thanks to the authors for carefully addressing the comments. However, my main concern over the scientific merit of the paper still remains: undifferentiated sedentary time.
I looked up the papers the author mentioned, particularly “The Association of Sedentary Behaviour and Cognitive Function in People Without Dementia: A Coordinated Analysis Across Five Cohort Studies from COSMIC”. The paper also emphasizes " specific types of sedentary behaviour may differentially influence cognition which should be investigated further”.
I don't think "occupational and leisure" captures the elements that are central to the investigation of cognitive performance. Playing video games in leisure time can contribute to creativity, which seems to outweigh doing boring, unengaging work. Is there a way you can specify the activity into executive domains: memory, attention, and cognitive flexibility? That way will help to move the field forward although it’s not well-conceptualized.
Author Response
Dear reviewer, we appreciate your observation. We are afraid there is no way we can specify the activity into executive domains once we do not have this data. As explained previously, we have a theoretical foundation for this paper, however, we understand your point of view and it seems fair to us. As we cannot analyze the specific SB, we can highlight this as a limitation of the study (please, find it in Lines 388-393). We will appreciate suggestions. "It is also important to highlight that we assess the total sitting time, without distinguishing activities performed while in the sitting position (e.g.: reading, writing, driving etc). Although previous studies also evaluated the total sitting time instead of specific activities [19], we encourage future studies to analyze specific types of sedentary behaviour, which seems to play a central role in cognitive performance assessment." Thank you for your time and contributions to this paper. BestReviewer 3 Report
Dear authors,
Thank you for clarifying the questions posed.
As regards question 7, I am of the opinion that this limitation should be mentioned.
Congratulations!
Author Response
Dear reviewer, we thank you for your recommendation. We brought it as a limitation. Please find it in Lines 384-388:
"In addition, we did not specify sedentary behaviour according to occupational status (i.e., retired or active). Although in this study 77.7% of the sample is active we consider that the SB of a person who works and a person who no longer works may differentially influence cognition, and it should be investigated in studies with a more representative sample of retirees."
Thank you for your time and contributions to this paper.
Best,